# SHDocs: A dataset, benchmark, and method to efficiently generate high-quality, real-world specular highlight data with near-perfect alignment

**Jovin Leong**
Home Team Science & Technology Agency
jovin_leong@htx.gov.sg

**Ming Di Koa**
Home Team Science & Technology Agency
koa_ming_di@htx.gov.sg

**Benjamin Cham**
Home Team Science & Technology Agency
benjamin_cham@htx.gov.sg

**Shaun Heng**
Home Team Science & Technology Agency
shaun_heng@htx.gov.sg

## Abstract

A frequent problem in vision-based reasoning tasks such as object detection and optical character recognition (OCR) is the persistence of specular highlights. Specular highlights appear as bright spots of glare that occur due to the concentrated reflection of light; these spots manifest as image artifacts which occlude computer vision models and are challenging to reconstruct. Despite this, specular highlight removal receives relatively little attention due to the difficulty of acquiring high-quality, real-world data. We introduce a method to generate specular highlight data with near-perfect alignment and present SHDocs—a dataset of specular highlights on document images created using our method. Through our benchmark, we demonstrate that our dataset enables us to surpass the performance of state-of-the-art specular highlight removal models and downstream OCR tasks. We release our dataset, code, and methods publicly to motivate further exploration of image enhancement for practical computer vision challenges.[1]

## 1 Introduction

Specular highlights are bright, localized reflections of light that appear as white spots or glare on reflective surfaces. These highlights naturally occur when the angle of reflection of light on a surface equals the angle of incidence, resulting in organized reflections of light that manifest as bright visual artifacts. These specular highlight artifacts result in image occlusion and are persistent problems in real-world computer vision tasks.

This occlusion is especially problematic in vision-based reasoning tasks such as optical character recognition (OCR) [29, 3, 22], object detection and recognition [32, 21], and unmanned vision-based systems [1, 39, 14] which necessitate a high degree of accuracy yet involve environments with high light intensity. As such, efforts have been made to develop image enhancement approaches to remove specular highlights before these images are used in downstream computer vision tasks [25, 13, 16, 3].

Existing works have highlighted the difficulty in developing specular highlight removal image enhancement models owing to the limited datasets available [20, 38, 22]. Due to the physical processes underlying specular highlights, it is challenging to generate aligned counterfactual images without complex experimental setups that allow researchers to vary the source and intensity of light. Meanwhile, the use of synthetic data has been explored to address this data scarcity with some success

---

[1] https://github.com/JovinLeong/SHDocs

38th Conference on Neural Information Processing Systems (NeurIPS 2024) Track on Datasets and Benchmarks.

[16, 9, 3, 22, 18]. However, the synthetic data approach has been observed to exhibit generalizability limitations when evaluated in real-world applications of computer vision [38, 20].

Table 1: Leading public specular highlight datasets

| Dataset | No. samples | Size | Real data? | Type |
|---------|-------------|------|------------|------|
| WHU-Specular Dataset [10] | 4310 image-mask pairs | 2.2 GB | Yes | Image masks |
| SHIQ [11] | 10825 scenes, 43300 images | 925.9 MB | Yes | Images in the wild |
| PSD [38] | 2210 scenes, 13380 images | 7.8 GB | Yes | Objects |
| SD1, SD2, RD [16] | 30000 images | 23.7 GB | 92% Synthetic | Text in the wild |
| SSHR [9] | 135000 images | 5.3 GB | Synthetic | Objects |
| SHDocs (Ours) | 3184 scenes, 19104 images | 13.5 GB | Yes | Documents |

This challenge motivated us to explore approaches to generate specular highlight data with neither the constraints of synthetic data nor the expensive, effort-intensive experimental setups prohibitive to many researchers. We developed a process leveraging polarized sensors and a polarized light setup to generate high-quality specular highlight data with near-perfect counterfactual alignment cheaply. We use our method to produce a dataset of real-world specular highlights on document images—a computer vision domain in which specular highlights are pertinent but lacking data. Sample images from our dataset, SHDocs, are shown in Figure 1.

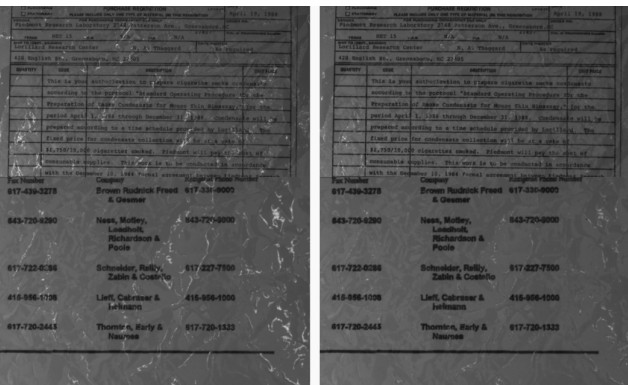

Figure 1: Sample document images from the SHDocs dataset with target images exhibiting specular highlights on the left, and deglared counterfactuals on the right

Our dataset is the only openly available real-world specular highlight dataset for document images; other leading datasets are shown in Table 1. Our dataset forms a benchmark which we use to assess leading specular highlight models and perform a generalizability study to evaluate how well our dataset generalizes across specular highlight tasks.

The salient contributions of our work are as follows:

1. The SHDocs dataset: A dataset of real specular highlights on document images based on the FUNSD document dataset [19] with ground truth annotations.

2. The source code and pipeline for generating aligned specular highlight image data with a Sony IMX250MZR, CMOS, 2/3" sensor.

3. A publicly released benchmark to evaluate specular highlight removal models in terms of image quality and OCR performance.

## 2 Related work

### 2.1 Specular highlight removal

Early works such as Guo et al. [13] and Gang et al. [12] leverage conventional computer vision methods for removing specular highlights. Fu et al. [11] introduce the SHIQ dataset and develop a model to detect and extract specular highlight masks from the Murmann et al. [27] multi-illumination images in the wild dataset. Fu et al. [10], Esfahani and Wang [7], Anwer et al. [2] focus on developing models to perform specular highlight detection by learning from specular highlight image masks.

Subsequent works focus on deep learning approaches to develop image enhancement models. Wu et al. [38] introduce a large specular highlight dataset by capturing images in fixed and random polarization angles which they use to train a Generative Adversarial Network (GAN) to remove specular highlights. Hou et al. [16] and Huang et al. [18] develop multi-stage models that detect and subsequently remove specular highlights. Fu et al. [9] propose a three-stage specular highlight removal network for highlight removal and subsequent enhancement. Finally, Hu et al. [17] introduce an adaptive highlight-aware module to develop a network that adaptively removes specular highlights.

A common pain point emerges from the specular highlight detection and removal literature: the scarcity of real-world specular highlight datasets. Dataset development efforts by Wu et al. [38], Fu et al. [11], and Hou et al. [16] have been significant but are high-effort and face limited scalability. Meanwhile, attempts to use synthetic data in modeling by Hou et al. [16], Chen et al. [3], Fu et al. [9] have demonstrated generalizability limitations when applied to real-world applications which manifest as hallucinations and poor real-world image enhancement outcomes.

### 2.2 Specular highlights in textual data

Existing works have sought to detect and remove specular highlights in textual image data through deep-learning image enhancement models. Rodin et al. [30] apply a lightweight convolutional neural network approach to detecting specular highlights on documents. Lahiri et al. [22] present a deep classification model to determine if document images have glare. Hou et al. [16] developed a text-aware two-stage network to detect and remove specular highlights in documents. Meanwhile, Chen et al. [3] develop a deep trident decomposition network for glare removal in license plates. Notably, the models developed by Hou et al. [16], Lahiri et al. [22], and Chen et al. [3] rely on synthetic data.

### 2.3 Polarization-based data collection methods

Polarization sensors are image sensors with integrated polarizers which we use in our study to collect aligned specular highlight data as detailed in Section 3.1. Several existing works in the literature sought to similarly use polarization methods to collect reflection and specular highlight data. Yang et al. [40] uses a Sony DFW-X70 camera and varies the polarization angles on polarization filters both on the camera and their light source to accumulate data which they use in their experimentation. Lei et al. [23] use a PHX050S-P polarization camera and panes of glass to create polarized reflection data which they use to develop a reflection removal image enhancement model. Wen et al. [37] generate specular highlight data by passing light through polarization filters which are rotated until there is no specular reflection captured by a separate polarization sensor.

Our method extends these works by devising an enclosed setup that minimizes unpolarized light with illumination through polarization filters at fixed angles and an algorithmic process to more readily generate specular highlight data with a focus on text recovery.

# 3 The SHDocs dataset

## 3.1 Overview of the setup

The dataset was collected using a FLIR Blackfly S camera equipped with a Sony IMX250MZR, CMOS, 2/3" polarization sensor (BFS-U3-51S5P) which captures 75 frames per second at 5.0 megapixels for a total image resolution of $2448 \times 2048$. The camera effectively captures greyscale $1224 \times 1024$ images at four polarization angles with a pixel size of $3.45\mu m$ for every single shot. The four angles are $i_0$, $i_{45}$, $i_{90}$, and $i_{135}$; they can be observed in Figures 3e to 3h respectively. We combine this capability with our polarized light setup to generate near-perfect counterfactual specular highlight images that form our SHDocs dataset.

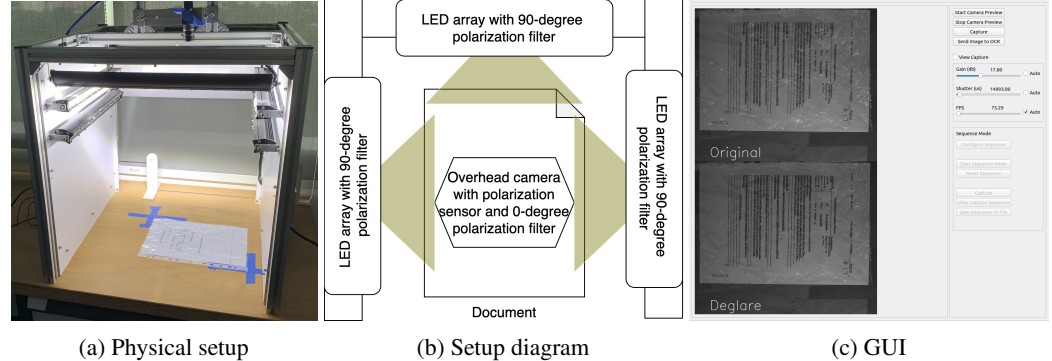

(a) Physical setup      (b) Setup diagram      (c) GUI

Figure 2: SHDocs data collection setup and GUI

We built an enclosure illuminated by light-emitting diode (LED) arrays attached to polarizing filters at fixed polarization angles as shown in Figure 2a. We fix the polarization angles because specular highlights generated by the polarized light from our setup are removed only if the polarization angle is perpendicular to any of the polarization directions of the polarization sensor. Figure 2b illustrates how the polarization filters are placed orthogonal to the polarization sensor.

An implication of this is that our method cannot filter out specular highlights from unpolarized light or polarized light at angles that are not orthogonal to our polarized sensor. This can be observed in Figures 1 and 3d where the deglared images still exhibit some specularity; this is a limitation of polarization filter methods. Thus, the enclosed setup was designed to limit the amount of unpolarized light entering the enclosure such that most of the light that generates the specular highlights would be polarized light that the polarization sensor can subsequently filter out.

## 3.2 Building the SHDocs dataset

We use the FUNSD dataset by Jaume et al. [19] as our base set of document data. The documents were printed and inserted into transparent filing pockets of differing quality and textures to generate specular highlights through their reflective surfaces. The FUNSD dataset comprises 199 fully annotated real document forms, 31485 words, 9707 semantic entities, and 5304 relations. We chose the FUNSD dataset as it was closely aligned with our experimental objective of creating realistic document data and is a widely used dataset for document analysis models and benchmarks [26, 8]. The transparent filing pockets simulate lamination and filing methods commonly used to waterproof documents and logistic labels. The texture and shape of these pockets refract the light at various angles and generate specular highlights under the polarized lights of our setup. We use a custom application with a graphical user interface (GUI) as shown in Figure 2c to facilitate image capture. The source code for the application is included in our publicly accessible GitHub repository.

For each of the 199 documents in the FUNSD, we took 15 images with different transparency layers and 1 unfiltered image without any transparency layers to create different specular highlight conditions for each document. For each image capture, our polarization sensor obtains 4 images corresponding to different polarization angles captured by the polarization sensor.

We further obtain 2 images by reconstructing the normalized Stokes parameter $S_0$ which is equivalent to a "normal" image that we might expect from a standard camera; and the "deglared" image through

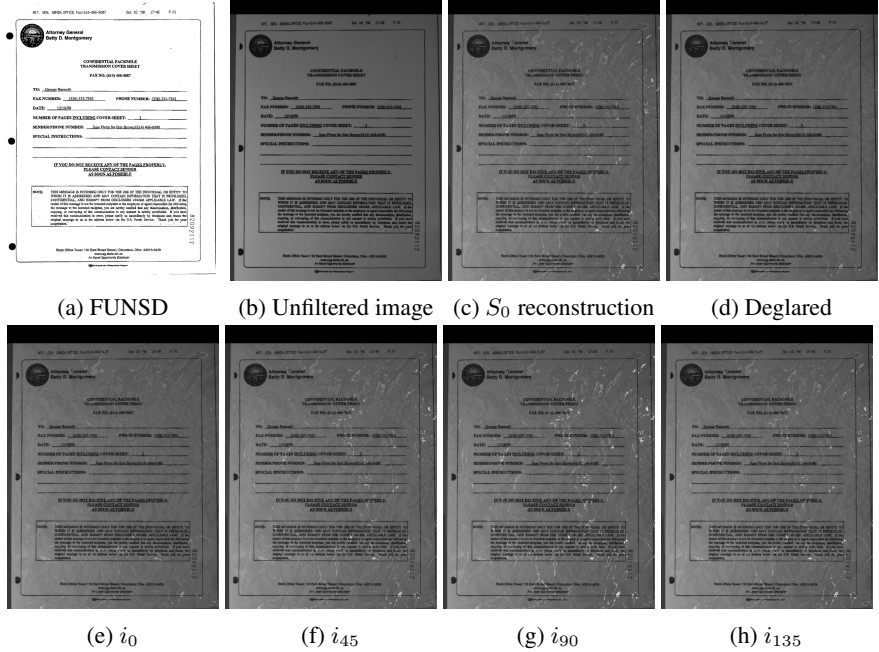

| (a) FUNSD | (b) Unfiltered image | (c) $S_0$ reconstruction | (d) Deglared |
| (e) $i_0$ | (f) $i_{45}$ | (g) $i_{90}$ | (h) $i_{135}$ |

Figure 3: Image captures and reconstructions

a glare removal process. The $S_0$ Stokes parameter is obtained by adding the intensities of the vertically and horizontally polarized pixels i.e. $S_0 = i_0 + i_{90}$. [34]. Meanwhile, the deglared image is obtained by taking the pixels with the lowest intensity across all 4 polarization angles i.e. we apply minimum pooling for each $2 \times 2$ matrix the polarization sensor returns for each pixel. These images represent the normal image with specular highlights and the counterfactual without specular highlights respectively. A diagram illustrating our complete data collection and process pipeline is included in the documentation available on the dataset's public code repository.

## 4    Experiments

Our experiments seek to benchmark leading specular highlight removal models on the SHDocs dataset to quantitatively assess how these models fare in terms of image enhancement and OCR performance. Through this, we seek a more complete understanding of the specular highlight removal space and hope to gauge the impact of our dataset and benchmark on image enhancement research.

### 4.1    Experimental procedure

Our benchmark consists of two phases. The first phase is a quantitative image quality assessment, where we pass the $S_0$ images from the SHDocs evaluation set through specular highlight models and evaluate the enhanced images with conventional quantitative image enhancement metrics using the deglared image, as described in Section 3.2, as the ground truth counterfactual. The second phase involves OCR evaluation of the enhanced image outputs, where we pass the enhanced images to OCR models for inference and then evaluate how the specular highlight removal models have impacted OCR text recovery outcomes. We also use Cho et al. [5]'s MIMO-UNetPlus model in our experiments. The MIMO-UNetPlus model employs a generic U-Net [31] architecture designed for image deblurring tasks and is not trained on specular highlight data. In conducting our experiments, we used Amazon Web Services' g4dn.4xlarge Nvidia T4 GPU-enabled virtual machines running on Deep Learning OSS Nvidia Driver Amazon Linux 2 Amazon Machine Image for PyTorch 1.13.1.[2]

---

[2]https://aws.amazon.com/ec2/instance-types/g4/

### 4.2 Quantitative image quality assessment

#### 4.2.1 Metrics

To quantitatively assess the image enhancement outcomes of specular highlight models, we adopt peak signal-to-noise ratio (PSNR) and structural similarity index measure (SSIM) as in Fu et al. [11], Wu et al. [38], Wen et al. [37]. We include universal image quality index (UIQI) as a further measure of image quality [35]. Our assessment compares $S_0$ images enhanced by the models to the deglared ground truth images. Higher PSNR, SSIM, and UIQI scores imply better image enhancement outcomes. We use Lightning AI's [6] implementations of the above metrics and report the average PSNR, SSIM, and UIQI of the enhanced images generated by models on the evaluation set.

#### 4.2.2 Results

The quantitative image quality assessment results comparing the enhanced $S_0$ images to the deglared ground truth images are shown in Table 2. Our findings indicate that the performance of specular highlight removal models on SHDocs is mixed. Except for M2-Net which had the best PSNR performance, all other specular highlight removal models fared worse than the baseline where no image enhancement had been applied. This result suggests that the image enhancements have tended to worsen image quality in terms of PSNR, SSIM, and UIQI relative to the deglared counterfactuals.

Table 2: SHDocs evaluation dataset

| Model | PSNR | SSIM | UIQI |
|---|---|---|---|
| No enhancement | 32.18 | **0.9589** | **0.8241** |
| Fu et al. [11] | 30.48 | 0.8599 | 0.5153 |
| M2-Net [18] | **32.72** | 0.9426 | 0.7201 |
| TSHRNet [9] | 30.66 | 0.9565 | 0.7693 |
| Hu et al. [17] | 30.96 | 0.9335 | 0.6874 |
| MIMO-UNetPlus [5] | 31.66 | 0.9492 | 0.7996 |

From Figure 4, visual observation of the $S_0$ target and the deglared ground truth reveal that the deglared image is an imperfect counterfactual for image quality assessment. Although the deglared image has less specularity than the $S_0$ image, it still contains specular highlights from unpolarized light sources that were not filtered out; this is discussed in Section 5.1. Consequently, certain specular highlight enhancements by Hu et al. [17] and M2-Net are erroneously marked down, negatively affecting the experimental results' metric performance and reliability. Additionally, the diffuse effects and downsampling performed by Fu et al. [11], Hu et al. [17] and TSHRNet have also diminished metric performance.

However, the literature has widely acknowledged the limitation of such quantitative image quality metrics [36] [41]. Even as the outputs of the above models score worse than the baseline without enhancement, their enhanced images may have greater qualitative visual appeal or usefulness. This motivates us to quantitatively assess how the enhanced images generated by these models impact the performance of OCR models in detecting and recognizing text within said images.

### 4.3 OCR performance

In evaluating the performance of OCR models on the enhanced image outputs of specular highlight models, our objective is not to directly assess how well OCR models perform in document processing as existing works have extensively explored [15, 4, 8]. Instead, we aim to use OCR performance to gauge the impact of specular highlight models in enhancing images for use in OCR. We restricted our models to Amazon Textract[3], EasyOCR[4], and Tesseract [33] as a representative sample of enterprise and open-source OCR models for document processing. We first determine how each OCR model performs with the original documents from the FUNSD evaluation set. Next, we assess OCR performance on the unenhanced SHDocs evaluation set without transparency filters followed by the performance on the unenhanced SHDocs evaluation set with transparency filters to serve as baselines. Finally, we pass the SHDocs evaluation set through specular highlight removal models and evaluate how the OCR models perform on the enhanced images.

---

[3]https://aws.amazon.com/textract/
[4]https://github.com/JaidedAI/EasyOCR

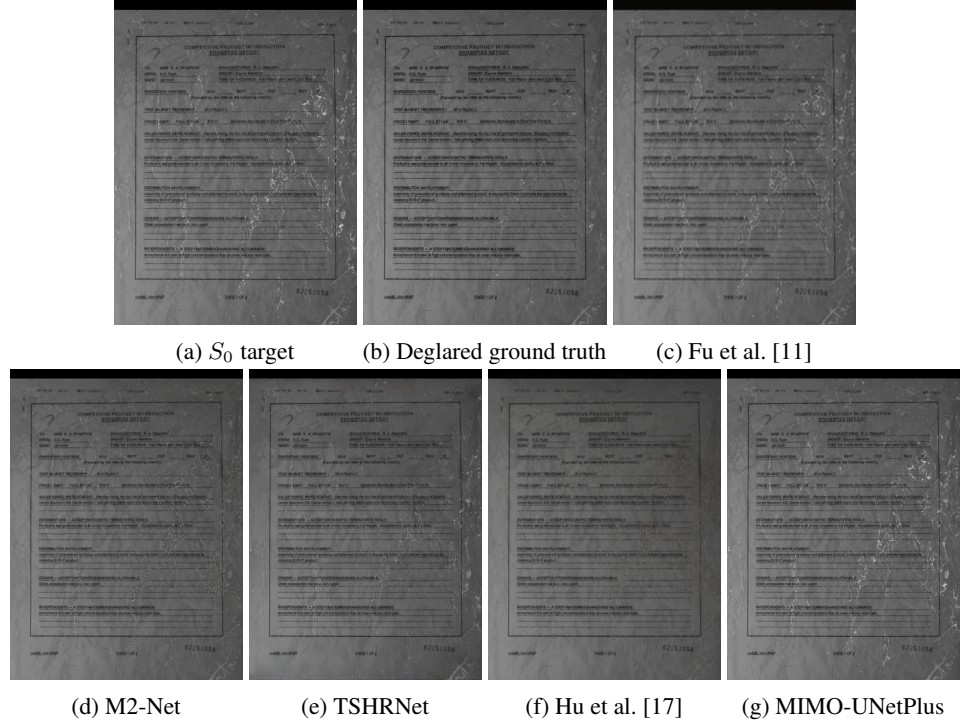

(a) $S_0$ target      (b) Deglared ground truth      (c) Fu et al. [11]

(d) M2-Net      (e) TSHRNet      (f) Hu et al. [17]      (g) MIMO-UNetPlus

Figure 4: Enhanced image quality assessment

### 4.3.1 Metrics

In our OCR performance assessment, we use three OCR evaluation metrics: Word Error Rate (WER), Character Error Rate (CER), and Levenshtein Edit Distance (LED) [24, 15] as implemented by Lightning AI's Torchmetrics library [6]. Lower WER, CER, and LED imply better OCR performance.

### 4.3.2 Results

The OCR evaluation results are shown in Table 3. The baseline results comparing the images with no filter and those with no enhancement suggest that the presence of specularity worsens OCR performance as per our hypothesis. Comparing the performance of specular highlight models, our results decisively indicate that MIMO-UNetPlus is the best model in terms of OCR performance on its enhanced images even though it has been untrained on specular highlight data. TSHRNet performs comparably across all OCR models—however, this only constitutes a marginal improvement over the baseline performance with no enhancement. Meanwhile, Huang et al. [18], Fu et al. [11], Hu et al. [17]'s models all performed worse than the baseline.

Table 3: OCR performance on SHDocs evaluation dataset

| | Textract | | | EasyOCR | | | Tesseract | | |
| Baseline | WER | CER | LED | WER | CER | LED | WER | CER | LED |
| --- | --- | --- | --- | --- | --- | --- | --- | --- | --- |
| Original FUNSD | 0.343 | 0.096 | 2.12 | 0.656 | 0.260 | 5.65 | 0.470 | 0.270 | 5.99 |
| No filter | 0.415 | 0.134 | 3.06 | 0.907 | 0.583 | 12.7 | 0.7314 | 0.586 | 12.9 |
| No enhancement | 0.593 | 0.358 | 7.89 | 0.950 | 0.723 | 15.8 | 0.846 | 0.686 | 15.1 |
| Model | | | | | | | | | |
| Fu et al. [11] | 0.960 | 0.865 | 18.8 | 1.00 | 0.998 | 21.6 | 1.00 | 0.960 | 20.8 |
| M2-Net [18] | 0.698 | 0.468 | 10.3 | 0.995 | 0.906 | 19.7 | 0.933 | 0.794 | 17.3 |
| TSHRNet [9] | 0.591 | 0.353 | 7.77 | 0.958 | 0.744 | 16.2 | 0.847 | 0.683 | 15.0 |
| Hu et al. [17] | 0.708 | 0.484 | 10.6 | 0.996 | 0.914 | 19.8 | 0.941 | 0.817 | 17.8 |
| MIMO-UNetPlus [5] | **0.582** | **0.342** | **7.53** | **0.942** | **0.686** | **15.0** | **0.824** | **0.653** | **14.3** |

Inspecting the image enhancement outputs from Fu et al. [11]'s model and M2-Net, we observe that the enhanced images tend to be blurry and low-resolution which likely impaired OCR performance. Additionally, the diffuse effects employed by Hu et al. [17]'s highlight removal network to reduce harsh specularity result in blurriness across affected areas which worsened the ability of OCR models to recognize textual character details. Meanwhile, the deblurring effect of MIMO-UNetPlus likely improved the ability of OCR models to recognize minute characters; despite the impactful specular highlight removal of models such as TSHRNet, the deblurring effect enabled MIMO-UNetPlus to achieve a better overall result. These results suggest that leading specular highlight removal models exhibit clear limitations in image enhancement domains such as textual data.

The SHDocs benchmark has enabled us to effectively discern between specular highlight removal models in terms of their ability to enhance images that OCR models subsequently consume. This has enabled us to identify gaps within the specular highlight removal space for textual images and illustrate limitations of image enhancement metrics such as UIQI, PSNR, and SSIM.

Altogether, our results in 4.2 and 4.3 demonstrate how performance in these image enhancement metrics did not translate to downstream image performance for OCR tasks. It is worth restating that the objective of this evaluation is not to provide a benchmark of the OCR models but to assess how image enhancement efforts affect OCR performance. We have not extensively verified that the FUNSD dataset was not used in the training of the above models—hence these results are not indicative of OCR model performance.

### 4.4 SHDocs generalizability study

To further evaluate the impact of SHDocs, we sought to study how SHDocs generalizes across specular highlight removal tasks through a two-part generalizability study. In the first part of our study, we selected two prominent specular highlight datasets: SHIQ [11] and PSD [38] and evaluated how leading specular highlight removal models, TSHRNet and Hu et al. [17], perform on these datasets using image enhancement metrics PSNR and SSIM. We retrain the generic U-Net model, MIMO-UNetPlus, on SHDocs and similarly evaluate how this model trained on SHDocs performs. Finally, as a baseline, we evaluate how a MIMO-UNetPlus deblurring model that has not been retrained on specular highlight data performs on these datasets.

In retraining MIMO-UNetPlus, we retained the original architecture and randomly initialized our model weights. We trained the model on the SHDocs training dataset with the MIMO-UNetPlus default hyperparameters: a batch size of 4, a learning rate of 0.0001, and a gamma of 0.5 for 3000 epochs with early stopping based on the validation PSNR.

Table 4: Generalizability of model trained on SHDocs across specular highlight datasets

| Model | Dataset trained on | SHIQ [11] | | PSD [38] | |
|---|---|---|---|---|---|
| | | PSNR | SSIM | PSNR | SSIM |
| TSHRNet [9] | SSHR, SHIQ, PSD [9, 11, 38] | 25.6 | 0.933 | 22.8 | 0.903 |
| Hu et al. [17] | SHIQ [11] | **33.9** | **0.980** | 27.5 | **0.970** |
| MIMO-UNetPlus [5] | SHDocs | 22.4 | 0.915 | **28.0** | 0.956 |
| MIMO-UNetPlus [5] | GoPro[28] | 23.1 | 0.903 | 18.1 | 0.705 |

The results in Table 4 indicate that although the MIMO-UNetPlus model retrained on SHDocs fares worse than other specular highlight removal models on the SHIQ dataset, it performs very comparably on PSD—with the highest PSNR on PSD's evaluation set. This is despite MIMO-UNetPlus having a model size of 64.6 MB compared to 468 MB and 412 MB in TSHRNet and Hu et al. [17] respectively. A detailed comparison of the model size is included in the public code repository.[5]

Additionally, comparing the performance of the base MIMO-UNetPlus and the MIMO-UNetPlus retrained on SHDocs, we observe that the retrained MIMO-UNetPlus largely outperforms the untrained MIMO-UNetPlus model, particularly on the PSD evaluation set. These results imply that the SHDocs dataset has been impactful in improving MIMO-UNetPlus' performance on specular highlight removal tasks and that it has enabled the retrained MIMO-UNetPlus model to perform competitively with other specular highlight removal models in leading specular highlight datasets.

---

[5]https://github.com/JovinLeong/SHDocs

In the second part of our study, we retrained the generic U-Net model architecture on various specular highlight datasets across different domains and hardware and evaluated the performance of each retrained model. This study sought to enable an apples-to-apples comparison to help assess the generalizability of SHDocs across specular highlight removal domains.

On top of SHDocs, we included the SHIQ [11] and RD [16] datasets as they contain specular highlight images in different domains and were captured with different hardware: SHIQ involves specular highlights on objects-in-the-wild while RD contains specularity on documents with Chinese characters. We used the MIMO-UNetPlus [5] architecture with the same training settings as before on all the datasets. As before, we include the original MIMO-UNetPlus [5] model trained on the GoPro dataset [28] in our evaluation as a baseline to compare against—since this dataset has been designed for image deblurring tasks and does not include any specular highlight data.

Table 5: MIMO-UNetPlus [5] model trained on specular highlight datasets

| Dataset trained on | SHIQ [11] | | RD [16] | | SHDocs | |
|---|---|---|---|---|---|---|
| | PSNR | SSIM | PSNR | SSIM | PSNR | SSIM |
| SHIQ [11] | **31.81** | **0.9569** | 16.10 | 0.7681 | 33.76 | 0.9258 |
| RD [16] | 17.79 | 0.8362 | **21.39** | **0.8407** | 20.85 | 0.8924 |
| SHDocs | 23.97 | 0.9104 | 16.12 | 0.7575 | **42.45** | **0.9692** |
| GoPro [28] | 22.52 | 0.8877 | 15.27 | 0.7381 | 31.61 | 0.9288 |
| No enhancement | 23.52 | 0.9240 | 15.37 | 0.7704 | 32.55 | 0.9445 |

The results in Table 5 show that, as expected, the MIMO-UNetPlus model performs best when it is evaluated on the same dataset on which it was trained. Notably, the results for the model trained on SHDocs in this study differ from our earlier generalizability study when evaluated on the SHIQ evaluation set. This difference is attributed to the random initialization of weights and the different early-stopping outcomes during training. The models in both generalizability studies are available for download on our public GitHub repository.[6]

We observe that the model trained on SHDocs is competitive with SHIQ and RD in all of the evaluation sets (discounting instances where the MIMO-UNetPlus model is evaluated on the dataset it is trained on). This is indicative that SHDocs exhibits a degree of generalizability to other specular highlight domains. Crucially, we find that the model trained on SHDocs outperforms the baseline model trained on the GoPro dataset in every evaluation set. This finding further suggests that SHDocs is impactful in generalizing across specular highlight domains.

Taken together, our findings from the above studies support the hypothesis that the SHDocs dataset is useful for image enhancement and exhibits generalizability across hardware and specular highlight removal tasks.

## 5   Discussion

In this paper, we have proposed an efficient method to generate high-quality specular highlight data, a specular highlight dataset comprising over 3000 document scenes with 19000 images, and a benchmark for evaluating image enhancement models through downstream OCR performance. Our experiments demonstrate limitations in existing methods of specular highlight removal for textual data and we provide the means to advance research in this domain. We believe that low-level innovations in the computer vision space as we have achieved shape how we approach vision-based reasoning and bolster more complex developments. We hope attention is paid to accessible methods of generating quality data to enable the broader community to advance research in machine learning and computer vision. We release our dataset and our code publicly to further efforts in this vein.

---

[6]https://github.com/JovinLeong/SHDocs/tree/main/model

### 5.1 Limitations

1. Our process, GUI, and pipelines have been designed for FLIR Blackfly S and Sony IMX250MZR. Although our work can be adapted to other sensors, this will be an obstacle for researchers with hardware access. Regardless, we view that our setup is reasonably specified as the components are commercially available and relatively inexpensive compared to conventional experimental setups required to capture specular highlight data.

2. Our method to generate and process specular highlights relies on specular highlights formed from polarized light. As such, our method cannot create alignment for specular highlights from unpolarized light or polarized light at angles that are not orthogonal to our polarized sensor. Failure cases can be observed in Figures 1 and 3d where the images deglared using our method still exhibit specularity. Although this effect can be mitigated by reducing unpolarized light (as we have done through our enclosed setup), this limitation of our method will inhibit the creation of datasets in conditions where light is largely unpolarized and limit the applicability of our method across contexts.

3. Our dataset and OCR evaluation only involve documents from FUNSD which exhibits a degree of style and context homogeneity in that the documents are from the United States of America and cover only the English language. This might result in generalization limitations when applied to documents with different contexts and languages. A future research direction could be to extend the dataset to cover a wider variety of contexts and languages through datasets such as HuggingFace's recently released pixparse PDF dataset.[7]

### 5.2 Social impact and ethical considerations

Our work extends the capabilities of researchers in the image enhancement and document analysis space. Such research will have applications in logistics, data processing, and administrative industries and may create obsoletion risks to occupations involving manual data entry. We view that these are consequences inherent to machine learning and automation that organizations and users must appropriately manage. Additionally, the hardware requirements highlighted in Section 5.1 can result in accessibility barriers for researchers seeking to replicate or extend our work. However, we have determined that our approach remains significantly more economical than the larger, controlled experimental setups employed in previous works and constitutes an overall increase in accessibility.

### 5.3 Usefulness of dataset and method

Through our work, we demonstrated that the SHDocs dataset is an insightful benchmark that highlights limitations in the specular highlight removal space—particularly for textual image data. We believe our benchmark constitutes a more complete assessment of image enhancement outcomes in this domain than quantitative image enhancement metrics as the inclusion of OCR metrics provides a more purposeful proxy for real-world image quality. Our benchmark enables the development of more impactful and practical image enhancement models that are better suited to textual images.

We are confident that SHDocs will benefit researchers tackling specularity in real-world images and documents and supports the development of text-aware image enhancement models. By including all unprocessed frames from 4 polarization angles, SHDocs also can be used in domains such as light modeling and simulation. Similarly, vision-foundation models can leverage our dataset to recognize and handle specularity. Finally, our method constitutes an efficient and low-effort method of generating and handling specular highlights. Our method can be used to form image datasets or in logistic applications that require illumination without specular highlights.

---

[7]https://huggingface.co/collections/pixparse/pdf-document-ocr-datasets-660701430b0346f97c4bc628

## Acknowledgments and Disclosure of Funding

This research was supported by the Sensemaking and Surveillance Centre of Expertise at the Home Team Science and Technology Agency (HTX), Singapore under the guidance of Yadong Wang, Mikhail Kennerley, Christopher Sia, and Kian Boon Lim. Additional support was provided by Tiong Kai Tan, and Soon Heng Ang from the Sensemaking and Surveillance Centre of Expertise along with Natasha Koh, Bee Ling Ng, and Gina Leow from the Chemical, Biological, Radiological, Nuclear, and Explosives Centre of Expertise.

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
