# A    Appendix

Include extra information in the appendix. This section will often be part of the supplemental material.
Please see the call on the NeurIPS website for links to additional guides on dataset publication.

1. Submission introducing new datasets must include the following in the supplementary materials:

   (a) Dataset documentation and intended uses. Recommended documentation frameworks include datasheets for datasets, dataset nutrition labels, data statements for NLP, and accountability frameworks.

   (b) URL to website/platform where the dataset/benchmark can be viewed and downloaded by the reviewers.

   (c) URL to Croissant metadata record documenting the dataset/benchmark available for viewing and downloading by the reviewers. You can create your Croissant metadata using e.g. the Python library available here: https://github.com/mlcommons/croissant

   (d) Author statement that they bear all responsibility in case of violation of rights, etc., and confirmation of the data license.

   (e) Hosting, licensing, and maintenance plan. The choice of hosting platform is yours, as long as you ensure access to the data (possibly through a curated interface) and will provide the necessary maintenance.

2. To ensure accessibility, the supplementary materials for datasets must include the following:

   (a) Links to access the dataset and its metadata. This can be hidden upon submission if the dataset is not yet publicly available but must be added in the camera-ready version. In select cases, e.g when the data can only be released at a later date, this can be added afterward. Simulation environments should link to (open source) code repositories.

   (b) The dataset itself should ideally use an open and widely used data format. Provide a detailed explanation on how the dataset can be read. For simulation environments, use existing frameworks or explain how they can be used.

   (c) Long-term preservation: It must be clear that the dataset will be available for a long time, either by uploading to a data repository or by explaining how the authors themselves will ensure this.

   (d) Explicit license: Authors must choose a license, ideally a CC license for datasets, or an open source license for code (e.g. RL environments).

   (e) Add structured metadata to a dataset's meta-data page using Web standards (like schema.org and DCAT): This allows it to be discovered and organized by anyone. If you use an existing data repository, this is often done automatically.

   (f) Highly recommended: a persistent dereferenceable identifier (e.g. a DOI minted by a data repository or a prefix on identifiers.org) for datasets, or a code repository (e.g. GitHub, GitLab,...) for code. If this is not possible or useful, please explain why.

3. For benchmarks, the supplementary materials must ensure that all results are easily reproducible. Where possible, use a reproducibility framework such as the ML reproducibility checklist, or otherwise guarantee that all results can be easily reproduced, i.e. all necessary datasets, code, and evaluation procedures must be accessible and documented.

4. For papers introducing best practices in creating or curating datasets and benchmarks, the above supplementary materials are not required.

## B   Supplementary materials

### B.1   Dataset details

#### B.1.1   Dataset documentation and intended uses

Dataset documentation is provided in our public public GitHub repository.[1]

Additional details on how data can be used and how users can utilize the SHDocs Croissant metadata are provided within the data directory of our public GitHub repository

#### B.1.2   URL to website/platform for dataset/benchmark

The dataset and benchmark are hosted on OneDrive. Download and usage instructions for our dataset and benchmark are detailed in our public GitHub repository

#### B.1.3   URL to Croissant metadata

The SHDocs Croissant metadata can be accessed on our public GitHub repository here.

Additional details on how data can be used and how users can utilize the SHDocs Croissant metadata are provided within the data directory of our public GitHub repository.

#### B.1.4   Statement on author responsibility

We, the undersigned authors, hereby declare that we collectively bear full and complete responsibility for the content of the work titled "SHDocs: A dataset, benchmark, and method to efficiently generate high-quality, real-world specular highlight data with near-perfect alignment". This includes—but is not limited to—ensuring that all data, materials, and content included in this work are original or appropriately cited and that we have obtained all necessary permissions and rights for any third-party materials used.

We affirm that our work complies with all applicable laws and regulations regarding intellectual property, copyright, and ethical standards. In the event of any dispute or violation of rights, we, the authors, will assume all liability and responsibility, and we will take all necessary steps to resolve any issues that may arise.

We acknowledge that The Conference and Workshop on Neural Information Processing Systems (NeurIPS) is not responsible for any content-related issues and that we indemnify and hold NeurIPS harmless against any claims, damages, or losses arising from the publication of this work.

Signed,

Jovin Wei Jie Leong
11 June 2024

Ming Di Koa
11 June 2024

Benjamin Wen Bin Cham
11 June 2024

Shaun Wei Quan Heng
11 June 2024

#### B.1.5   Hosting, licensing, and maintenance plan

The dataset will be hosted on enterprise OneDrive and Google Drive where the links will both be publicly available.

---

[1]https://github.com/JovinLeong/SHDocs

The data and code are licensed with The MIT License and the licensing details have been included within the public repository.

Dataset maintenance will be carried out by all authors; we will be actively working with the dataset for future projects and will use our findings to perform remediation when necessary. We will monitor and address all dataset issues raised on GitHub to ensure that the dataset remains accessible and usable.

Code maintenance will similarly be carried out by the authors based on our independent remediation and issues raised by users on GitHub. However, our code maintenance will only be on a best-effort basis—particularly when resolving issues arising from dependencies, development environments, and infrastructure.

## B.2 Accessibility

### B.2.1 Links to access the dataset and metadata

The link to access our data and our metadata are included in our public GitHub repository. Additionally, SHDocs can be downloaded from the following links:

SHDocs raw data: Microsoft OneDrive or Google Drive

SHDocs processed data: Microsoft OneDrive or Google Drive

### B.2.2 How the dataset can be read

The dataset can be most conveniently read using Croissant to obtain the dataset records in a standardized fashion. Users can download the dataset and use the provided Croissant metadata file to load the dataset records. Detailed instructions are provided within the data directory of our public GitHub repository

Alternatively, users can manually unzip the dataset and access the data directly. The dataset consists solely of .PNG images and JSON files; these can easily be read by standard libraries available in most programming languages.

### B.2.3 Long-term preservation

The dataset will be hosted on enterprise OneDrive and Google Drive where the links will both be publicly available.

Dataset maintenance will be carried out by all authors; we will be actively working with the dataset for future projects and will use our findings to perform remediation when necessary. We will monitor and address all dataset issues raised on GitHub to ensure that the dataset remains accessible and usable.

Additionally, we will explore the use of data repositories such as Hugging Face Hub later on to maximize data preservation.

### B.2.4 Explicit license

The data and code are licensed with The MIT License and the licensing details have been included within the public repository.

### B.2.5 Structured metadata

The SHDocs Croissant metadata can be accessed on our public GitHub repository here. Croissant's metadata structure is based on schema.org and is thus compliant to Web standards.

### B.2.6 Dereferenceable identifier

The SHDocs uses a public GitHub repository whose unique URL serves as its persistent identifier.

## B.3 Reproducibility

To ensure the reproducibility of our benchmark results, we have included everything needed to replicate our findings. This includes our trained model, which is part of the supplementary materials. We've also made all the code used in our experiments publicly accessible through our public GitHub repository. The repository contains all necessary scripts, configuration files, and dependency information.

We've provided detailed documentation within the repository to guide you through the entire process. This documentation covers how to set up the environment, run the code, and follow the exact steps we used for model training, evaluation, and result generation.

All external models and datasets by other authors that we used in our benchmark are clearly cited and publicly available.

Where possible, we have followed The Machine Learning Reproducibility Checklist v2.0 to ensure our experiments are documented and reproducible. By providing all these resources, we aim to make it straightforward for anyone to verify and build upon our work.