# OpenReview forum: "SHDocs: A dataset, benchmark, and method to efficiently generate high-quality, real-world specular highlight data with near-perfect alignment"
_NeurIPS.cc/2024/Datasets_and_Benchmarks_Track — NeurIPS 2024 Track Datasets and Benchmarks Poster_

### Official Review · Reviewer_Ka9r · 2024-07-18
**A dataset of real specular highlights on document images, and pipeline & code for generating aligned specular highlight image data**

**Rating:** 6
**Confidence:** 4
**Correctness:** The dataset is constructed in a sound…
**Clarity:** The paper is well written.

**Review:**

The SHDocs dataset and benchmark provide valuable resources for advancing research in specular highlight removal and enhancing image quality for OCR tasks. The proposed method offers an efficient and low-cost approach to generating aligned specular highlight data, addressing the scarcity of real-world datasets in this domain.

(1) Quality: The specular highlight data was captured by the camera, however, the ground-truth image still contains specular highlights.

(2) Clarity: The paper is well written.

(3) Originality & Significance: This work presents a new method to generate specular highlight data. The dataset fills the gap in the specular highlights on document images.

**Strengths:**

(1) The proposed method is capable of generating high-quality specular highlight data cheaply.
(2) The dataset fills the gap in the specular highlights on document images.
(3) The method, dataset, and evaluation code are open-sourced.

**Additional Feedback:**

None

**Documentation:**

This dataset provides details on data collection and organization, availability and maintenance, and ethical and responsible use.

**Ethics:**

There are no ethics concerns

**Limitations:**

The authors discuss some of the limitations in Sec. 5.1

**Opportunities For Improvement:**

(1) As shown in Fig. 3 (d) and mentioned in Sec. 4.2.2, the deglared image still contains specular highlights from unpolarized light sources that were not filtered out, is it indicates that the ground-truth image generated by the method is sub-optimal?

(2) This dataset was generated solely on the FUNSD dataset, which is homogenous in style.

(3) There is a lack of benchmark experiments that trained and then evaluated on the SHDocs dataset.

(4) In Sec. 4.4, the authors did not conduct experiments on the SD1, SD2, RD [16] dataset, which is a text-related dataset.

**Relation To Prior Work:**

This paper clearly discussed how this work differs from previous contributions

**Summary And Contributions:**

This paper introduces a method to generate specular highlight data with near-perfect alignment and presents SHDocs—a real dataset of specular highlights on document images created using the method. The SHDocs dataset comprises over 3000 document scenes with 19000 images and is the only openly available real-world specular highlight dataset for document images. Additionally, this paper proposes a benchmark for evaluating specular highlight removal models in terms of image quality and OCR performance.

---

> ### Author Rebuttal · Authors · 2024-08-15
>
> Thank you for your time and your review, we very much appreciate your feedback.
>
> > As shown in Fig. 3 (d) and mentioned in Sec. 4.2.2, the deglared image still contains specular highlights from unpolarized light sources that were not filtered out, is it indicates that the ground-truth image generated by the method is sub-optimal?
>
> Yes, this is a limitation of our method that we discuss in Section 5.1 (page 9, lines 280-283) and is a known limitation of all polarization-based methods of specularity filtering. The specular highlights observed in the figure can largely be attributed to unpolarized light introduced by our laboratory conditions and surface reflections within our set-up.
>
> These result in suboptimal conditions for dataset collection but can be mitigated by minimizing reflections through the use of dark surfaces within the setup and limiting the amount of unpolarized light entering the setup.
>
> Although suboptimal, we contend that SHDocs is sufficiently impactful in specular highlight removal tasks, as supported by our experiments and generalizability study—though future efforts by researchers and the broader community will stand to benefit through an awareness of these limitations.
>
> Thus, your feedback is useful in bringing to our attention the need for greater emphasis on our method’s limitations and how such limitations can be mitigated for researchers hoping to adopt our approach.
>
> In our revision, we have updated Section 5.1 (page 9, lines 280-283) to elaborate on the limitation of our method further and make reference to sample images in Figure 3(d) and Figure 1. We also include recommendations on how this limitation can be mitigated.
>
> > This dataset was generated solely on the FUNSD dataset, which is homogenous in style.
>
> Thank you for highlighting this. Yes, this is a known limitation of SHDocs that we sought to address in Section 5.1 (page 9, lines 284-287) where we acknowledged our source dataset (FUNSD) exhibits language and context biases.
>
> We selected the FUNSD dataset due to its prominence in the OCR space and relevance to our initial research work. However, we acknowledge this limitation and view that an extension of this work could expand the diversity of SHDocs to capture a broad variety of documents. This could include using the newly released PixParse dataset Hugging Face which contains a significantly larger corpus of documents.
>
> In our revision, we will incorporate your feedback to specify the style homogeneity of FUNSD in Section 5.1 (page 9, lines 284-287) and more constructively reference PixParse as a proposed alternative base dataset for those looking to extend our work.
>
> > There is a lack of benchmark experiments that trained and then evaluated on the SHDocs dataset.
>
> This is good feedback that has been duly noted. Our initial generalizability study sought only to evaluate the impact of SHDocs on specular highlight removal tasks beyond the textual domain.
>
> Hence, we evaluated the models only on other specular highlight datasets because we wanted to determine if SHDocs as a training dataset could enable a generic UNet model to learn enough about specularity to become competitive at specular highlight removal tasks by having models trained with different datasets evaluated on the same evaluation set.
>
> However, we acknowledge that the inclusion of experiments that train and evaluate the SHDocs dataset can provide a more complete assessment of SHDocs.
>
> In our latest revision based on your feedback, we updated our generalizability study in Section 4.4 (page 8, lines 239-263) to include an experiment involving the generic UNet model trained on different datasets—i.e. keeping the model constant but varying the training sets. Subsequently, we will evaluate the models on the evaluation sets of all the specular highlight datasets in consideration (including SHDocs). Through this, we will have a more complete assessment of the generalizability of SHDocs.
>
> We have attached the revised table containing experiments that train and evaluate on the SHDocs dataset with references in our rebuttal as “Table 1” in the “Rebuttal_Appendix.pdf.
>
> > In Sec. 4.4, the authors did not conduct experiments on the SD1, SD2, RD [16] dataset, which is a text-related dataset.
>
> This is a good point. Section 4.4 references our generalizability study where we sought to evaluate if SHDocs generalizes to other specular highlight domains beyond text (page 8, lines 239-240). As SD1, SD2, and RD are text-related, our initial perspective was that the inclusion of those datasets would not give us a strong sense of how well SHDocs generalizes to other forms of specularity such as images-in-the-wild.
>
> However, given your feedback and further consideration of how RD’s contain Chinese characters which would constitute a significant domain shift from SHDocs, we have revised our generalizability study in Section 4.4 (page 8, lines 239-263), as mentioned in our response to your earlier feedback, to train and evaluate a specular highlight model against RD. However, we contend that SD1 and SD2 would not constructively contribute to our study as they are highly synthetic datasets which will not make for a meaningful comparison—given that our objective is to develop and assess datasets with real-world applicability.
>
> We have attached the revised table which includes experiments involving the RD dataset with references in our rebuttal as “Table 1” in the “Rebuttal_Appendix.pdf.
>
> Thank you once again for your time and thoughtful review; please let us know if there are further areas that we can address.

---

> > ### Comment · Reviewer_Ka9r · 2024-08-30
> >
> > After reviewing the rebuttal and considering the feedback from other reviewers, I am inclined to maintain my original score.

---

> > > ### Author Response · Authors · 2024-08-31
> > > **Official comment by authors**
> > >
> > > Understood; thank you again for the feedback and the time.

---

### Official Review · Reviewer_uNsq · 2024-07-23
**Limitations of the SHDocs dataset**

**Rating:** 6
**Confidence:** 2
**Correctness:** Yes
**Clarity:** Yes

**Review:**

Overall, I believe the proposed specular highlight datasets will benefit the community. However, I have a few concerns.

1. In Table 4, the authors conduct ablation studies on the model generalization capabilities. However, the training datasets for these models are not uniform. I wonder if the authors could evaluate the performance under the same training settings. This would benefit the community.

2. Failure cases of models trained on the proposed datasets should be shown. It would be really helpful if the authors could provide some failure cases, as these would help researchers identify significant issues in this community.

**Strengths:**

1. This paper proposes a large-scale specular highlight dataset.

2. Experiments show the effectiveness of the proposed dataset.

**Additional Feedback:**

Please see the review part.

**Documentation:**

Yes

**Limitations:**

Yes

**Opportunities For Improvement:**

1. More failure cases should be shown.

2. The generalization evaluation is very important. My suggestion is that the authors conduct experiments using the same training dataset.

**Relation To Prior Work:**

Yes

**Summary And Contributions:**

This paper proposes a new method to generate specular highlight data with near-perfect alignment and presents SHDocs—a dataset of specular highlights on document images created using this method. Extensive experiments show that the model trained on the proposed dataset can surpass the performance of state-of-the-art specular highlight removal models and improve downstream OCR tasks.

---

> ### Author Rebuttal · Authors · 2024-08-15
>
> Thank you for your time and your review, we very much appreciate your feedback.
>
> > However, the training datasets for these models are not uniform. I wonder if the authors could evaluate the performance under the same training settings. This would benefit the community.
>
> This point is well acknowledged. Our generalizability study captured in Table 4 sought to evaluate the impact of SHDocs in specular highlight removal tasks such as specular highlights removal for images-in-the-wild in the SHIQ and PSD datasets.
>
> We demonstrated SHDocs’ generalizability by training a generic UNet model architected for deblurring tasks on SHDocs and comparing the performance of this resultant model against leading specular highlight removal models on the SHIQ and PSD evaluation sets.
>
> Hence, we evaluated the models under different training settings precisely because we wanted to determine if SHDocs as a training dataset could enable a generic UNet model to learn enough about specularity to become competitive at specular highlight removal tasks.
>
> However, through your feedback, we acknowledge that our generalizability study can be better expressed.
>
> In our latest revision based on your feedback, we updated our generalizability study in Section 4.4 (page 8, lines 239-263) to include an experiment involving the generic UNet model trained on different datasets—i.e. keeping the model constant but varying the training sets for a more apples-to-apples comparison. Subsequently, we evaluate the models on the evaluation sets of all the specular highlight datasets in consideration. Through this, we will have a more uniform experiment to evaluate the impact of SHDocs across specular highlight removal domains. We have attached the revised table with references in our rebuttal as “Table 1” in the “Rebuttal_Appendix.pdf.
>
> We will retain the existing findings of our generalizability study as we view that these insights are still useful for the community to understand the impact and limitations of SHDocs.
>
> > Failure cases of models trained on the proposed datasets should be shown. It would be really helpful if the authors could provide some failure cases, as these would help researchers identify significant issues in this community.
>
> This feedback is duly noted. In our work, we discuss the limitations and failure cases of our contributions as follows:
>
> The limitations and failure cases of our data collection method are expressed in Section 5.1 (page 9, lines 280-283), where we discuss instances where our polarization-based method will fail to filter out specular highlights. This is illustrated in Figure 4(b).
> The limitations and failure cases of SHDocs (our dataset) are expressed in section 4.2.2 (page 6, lines 182-188) where we note that several of the counterfactuals are imperfect in that although they are aligned to the specular highlight images S_0 images, these counterfactuals still exhibit some specularity due to the aforementioned limitations in our method. Similarly, this is illustrated in Figure 4(b).
> We illustrate a failure case of our generic UNet model trained on SHDocs in Figure 4 where the enhanced image produced by the model performs visually worse than other models such as TSHRNet and M2-Net—though this was not explicitly elaborated upon in our writing.
>
> Nevertheless, your feedback helps us to recognize how our work benefits from a greater emphasis on these failure cases—such that the community can be more cognizant of the limitations of our contributions.
>
> In our revision, we updated our writing in Section 3.1 (page 4, lines 114-116) to emphasize the limitations of our method with clearer references to the examples in Figure 1 and Figure 4(c). We also updated the writing in Section 5.1 (page 9, lines 280-283) to highlight how the removal of specularity is imperfect—even if the alignment of images to their counterfactuals is perfect.
>
>
> Thank you once again for your time and thoughtful review; please let us know if there are further areas that we can address.

---

### Official Review · Reviewer_FWs1 · 2024-07-25
**SHDocs: A dataset, benchmark, and method to efficiently generate high-quality, real-world specular highlight data with near-perfect alignment**

**Rating:** 7
**Confidence:** 3
**Correctness:** Yes
**Clarity:** Yes

**Review:**

Pros:

- This paper presents a solid dataset for real-world specular highlight data for document images. The processed hardware set up is well-explained and the details are covered. Although I am not direclty working in this area, I believe this open-source dataset would benefit research along this line.
- The experiment section covers good insights and details on the limitation of existing metrics, how the removal could be evaluated by both image quality metric and OCR performance.

Cons:

- the generalization studies have been conducted across specular highlight removal tasks such as specular highlight removal in objects. However, in low-level tasks like this, the hardware/lens would produce nuance difference that might affect generalization. Since this work only leverages one type of hardware (camera/polarization filter/etc.), it would great to show it has no issue generalizing other hardwares to further prove its generalization.
- redundant citation in table 3 (Fu et al. [11] [11])

**Strengths:**

as mentioned above

**Additional Feedback:**

N/A

**Documentation:**

N/A

**Opportunities For Improvement:**

as mentioned above

**Relation To Prior Work:**

Yes

**Summary And Contributions:**

This paper addresses the issue of specular highlights in vision-based reasoning tasks by introducing a method to generate high-quality specular highlight data and presenting SHDocs, a dataset of document images with specular highlights. Their benchmark demonstrates that using SHDocs improves the performance of specular highlight removal models and OCR tasks.

---

> ### Author Rebuttal · Authors · 2024-08-15
>
> Thank you for your time and your review, we very much appreciate your feedback.
>
> > redundant citation in table 3 (Fu et al. [11] [11])
>
> Thank you for bringing this to our attention—the citation has been removed.
>
> > However, in low-level tasks like this, the hardware/lens would produce nuance difference that might affect generalization. Since this work only leverages one type of hardware (camera/polarization filter/etc.), it would great to show it has no issue generalizing other hardwares to further prove its generalization.
>
> This comment is correct in highlighting how the dataset exhibits a hardware bias. We acknowledge the significance of this bias and have emphasized the hardware used in collecting SHDocs (our dataset) in Sections 1 (page 2, line 57), 3.1 (page 3, lines 103-104), and 5.1 (page 9, lines 275-276).
>
> However, we contend that our work sought to show generalizability to other hardware through the generalizability study, where we used SHDocs to train a generic UNet architecture (originally designed for image deblurring tasks). We took this trained model and evaluated it against other specular highlight datasets which contain specular highlight data captured on different hardware. Our generalizability study showed that even though we only used a generic image enhancement model architected for image deblurring, we achieved competitive performance compared to leading specular highlight removal models. This implies that our dataset is able to exhibit generalizability across specular highlight removal tasks, as it enabled the generic UNet model to perform comparably to leading models on datasets captured using different hardware.
>
> Having said that, through your feedback, we acknowledge that our generalizability study can be better expressed to convey this sentiment.
>
> In our latest revision based on your feedback, we have removed the redundant citation and we have updated our writing in our generalizability study in Section 4.4 (page 8, lines 239-240) to specify that the evaluation of other datasets captured in different conditions and sensors enables us to assess the hardware generalizability of SHDocs.
>
> Thank you once again for your time and thoughtful review; please let us know if there are further areas that we can address.

---

### Comment · Area_Chair_k1fM · 2024-08-31
**Feedback on rebuttal**

Dear Reviewers, Thank you for your reviews. Please review the author rebuttals and give feedback as the deadline is tomorrow for the interactions. Thank you. Best, AC

---

### Decision · Program_Chairs · 2024-09-26

**Decision:**

Accept (Poster)

**Comment:**

Since both AC and SAC are non-responsive, PCs decide to give accept based on current reviewers' comments.